# Updating estimates of *Plasmodium knowlesi* malaria risk in response to changing land use patterns across Southeast Asia

Ruarai J. Tobin[1]*, Lucinda E. Harrison[2], Meg K. Tully[1], Inke N. D. Lubis[3], Rintis Noviyanti[4], Nicholas M. Anstey[5], Giri S. Rajahram[6], Matthew J. Grigg[5], Jennifer A. Flegg[2], David J. Price[1,7], Freya M. Shearer[1,8]*

**1** Infectious Disease Dynamics Unit, Melbourne School of Population and Global Health, The University of Melbourne, Melbourne, Australia, **2** School of Mathematics and Statistics, The University of Melbourne, Melbourne, Australia, **3** Department of Paediatrics, Faculty of Medicine, Universitas Sumatera Utara, Medan, Indonesia, **4** Eijkman Research Center for Molecular Biology, BRIN, Jakarta, Indonesia, **5** Menzies School of Health Research and Charles Darwin University, Darwin, Australia, **6** Infectious Diseases Society Kota Kinabalu Sabah, Menzies School of Health Research, Clinical Research Unit, Hospital Queen Elizabeth II, and Clinical Research Centre, Queen Elizabeth Hospital, Ministry of Health, Kota Kinabalu, Malaysia, **7** Doherty Institute for Infection and Immunity, The Royal Melbourne Hospital and The University of Melbourne, Melbourne, Australia, **8** Infectious Disease Ecology and Modelling Group, Telethon Kids Institute, Perth, Australia

* ruarai.tobin@unimelb.edu.au (RJT); freya.shearer@unimelb.edu.au (FMS)

**Data Availability Statement:** Prediction results for each bootstrapped model, rasters of summary statistics, the code used to produce results, and

## Abstract

### Background

*Plasmodium knowlesi* is a zoonotic parasite that causes malaria in humans. The pathogen has a natural host reservoir in certain macaque species and is transmitted to humans via mosquitoes of the *Anopheles* Leucosphyrus Group. The risk of human *P. knowlesi* infection varies across Southeast Asia and is dependent upon environmental factors. Understanding this geographic variation in risk is important both for enabling appropriate diagnosis and treatment of the disease and for improving the planning and evaluation of malaria elimination. However, the data available on *P. knowlesi* occurrence are biased towards regions with greater surveillance and sampling effort. Predicting the spatial variation in risk of *P. knowlesi* malaria requires methods that can both incorporate environmental risk factors and account for spatial bias in detection.

### Methods & results

We extend and apply an environmental niche modelling framework as implemented by a previous mapping study of *P. knowlesi* transmission risk which included data up to 2015. We reviewed the literature from October 2015 through to March 2020 and identified 264 new records of *P. knowlesi*, with a total of 524 occurrences included in the current study following consolidation with the 2015 study. The modelling framework used in the 2015 study was extended, with changes including the addition of new covariates to capture the effect of deforestation and urbanisation on *P. knowlesi* transmission.

the updated occurrence database have been made available at osf.io/k5bsa (DOI 10.17605/OSF.IO/K5BSA).

**Funding:** This study was supported through funding provided by the Australian Centre for International Agricultural Research (ACIAR), as part of the 'Evaluating zoonotic malaria transmission and agricultural and forestry land use in Indonesia' (ZOOMAL) project (LS/2019/116, www.aciar.gov.au). Further support for this project was provided by the National Health and Medical Research Council of Australia through its Centres of Research Excellence (ACREME, GNT1134989 www.nhmrc.gov.au). FMS was supported by the National Health and Medical Research Council of Australia Investigator Grant Scheme (Emerging Leader Fellowship, 2021/GNT2010051 www.nhmrc.gov.au). JAF was supported by the Australian Research Council (ARC, FT210100034 and DP200100747 www.arc.gov.au). LEH was supported by a Melbourne Research Scholarship from the University of Melbourne (www.unimelb.edu.au). MJG was supported by the National Health and Medical Research Council of Australia Investigator Grant Scheme (Emerging Leader 2 Fellowship, 2023/GNT2017436 www.nhmrc.gov.au). GSR and MJG were supported by the National Institutes of Health, USA (R01AI160457-01 www.nih.gov). GSR was also supported by the Malaysian Ministry of Health (Grant Number BP00500/117/1002 www.moh.gov.my). The funders had no role in study design, data collection and analysis, decision to publish, or preparation of the manuscript.

**Competing interests:** The authors have declared that no competing interests exist.

## Discussion

Our map of *P. knowlesi* relative transmission suitability estimates that the risk posed by the pathogen is highest in Malaysia and Indonesia, with localised areas of high risk also predicted in the Greater Mekong Subregion, The Philippines and Northeast India. These results highlight areas of priority for *P. knowlesi* surveillance and prospective sampling to address the challenge the disease poses to malaria elimination planning.

## Author summary

*Plasmodium knowlesi* is a parasite that can cause malaria when it infects humans. Although most people do not experience severe illness from *Plasmodium knowlesi* infection, a small number will develop serious or even fatal disease. The parasite is found naturally in some monkeys throughout Southeast Asia, and spreads from these monkeys to humans through mosquitoes. Previous research predicted where the risk of being infected is highest according to what we know about the environment across Southeast Asia, such as if there are forests in an area or if the altitude is high. In this work, we extend this previous research with more up-to-date data on environmental conditions and infections to predict the risk of being infected with *Plasmodium knowlesi*. We show that the risk *Plasmodium knowlesi* poses to humans is high across much of Southeast Asia, and that the disease will continue to challenge national goals to eliminate malaria.

## Introduction

*Plasmodium knowlesi* is a zoonotic pathogen of growing public health concern in Southeast Asia. The pathogen has a reservoir in *Macaca fascicularis* and the closely related *Macaca nemestrina* and *Macaca leonina* macaques, and is transmitted between macaques and from macaques to humans via mosquito vectors of the *Anopheles* Leucosphyrus Group [1,2]. Although demonstrated experimentally [3], evidence of direct human-to-human transmission of *P. knowlesi* occurring in nature is limited [1,4–7].

Infection by *P. knowlesi* most often causes mild to moderate illness in humans [8]. However, a range of outcomes are possible, with both asymptomatic infection [9–11] and severe disease being reported. Studies of patients presenting to health care facilities in Malaysia reported severe disease in around 6–9% of patients [12,13].

For humans, the likelihood of contracting a *P. knowlesi* infection has been found to be dependent upon a range of risk factors, with case-control and seroprevalence studies demonstrating associations between environmental variables and the occurrence of infection. A seroprevalence survey performed in Malaysia and the Philippines found that prior infection with *P. knowlesi* was associated with the proximity of forested areas to an individual's home and the clearing of forest near their home [14]. A similar study performed in northern Sabah, Malaysia, found associations between prior infection and an individual reporting that they have had activity in forested areas or that they have had contact with macaques [15]. A population-based case-control study performed within Sabah, Malaysia, found an association between current *P. knowlesi* infection and an individual reporting either that they had recently cleared vegetation or that their home was in proximity to long grass [11].

The spatial epidemiology of *P. knowlesi* malaria has historically been poorly understood. This is partially due to widespread misdiagnosis. By clinical presentation, the symptoms of *P.*

*knowlesi* infection can be easily misattributed to other major human species of malaria such as *P. vivax* or *P. falciparum* [16]. Under microscopic examination, the parasite appears almost identical to *P. malariae* [17] and the early ring stages of *P. falciparum* [18]. One review of historical microscopy diagnoses demonstrated that across 375 studies, 57% of *P. knowlesi* infections were misdiagnosed [17]. In addition to misdiagnosis, the understood spatial distribution of *P. knowlesi* malaria has been biased by differences in surveillance effort. Within peer-reviewed literature, reported *P. knowlesi* infections are most common in Malaysia, which is likely reflective of both high burden and a substantial surveillance effort in the country [19]. Indigenous cases of *P. knowlesi* malaria have also been detected in Brunei, Cambodia, Indonesia, Laos, Myanmar, the Philippines, Thailand, and Vietnam, but these have historically been the result of small-scale prospective sampling efforts and individual case reports. One study reported identifying *P. knowlesi* in India within the Andaman and Nicobar Islands [20].

The elimination of malaria in at least 20 countries by 2025 is listed as a key milestone of the World Health Organisation's *2016–2030 Global technical strategy for malaria* [21]. *P. knowlesi* presents a challenge to these efforts, since interventions that are effective against the human malaria species such as indoor residual spraying will be less effective against *P. knowlesi* due to the pathogen's persistence in wildlife reservoirs. Furthermore, cross-reactivity of antibodies between *P. knowlesi* and the closely genetically related *P. vivax* may provide protection against *P. knowlesi* infection [22], implying that the elimination of *P. vivax* in a region could lead to reduced immunity and subsequently an increase in the number of *P. knowlesi* infections [19].

The incidence of *P. knowlesi* in humans appears to be increasing within Southeast Asia; in Malaysia, the number of recorded human *P. knowlesi* infections doubled over the period from 2015 to 2018 [23]. A similar trend is visible in the rising number of case reports within Indonesia [24]. Though these trends may simply reflect improvements in surveillance [23], it has been suggested that deforestation in the region may be leading to a real increase in the number of human *P. knowlesi* infections [24–27]. A primary driver of deforestation in the region is the development of oil palm or timber plantations, which produce an environment that is believed to be of enhanced risk for *P. knowlesi* infection, with plantation labourers being required to live and work in proximity to recently disturbed forests that may contain *P. knowlesi* reservoirs and vectors.

Sustained transmission of vector-borne zoonoses can only occur at the nidus where pathogen, host and vector are present in sufficient abundance [28]. For each of these, certain constraints limit their distribution, for example: a pathogen may be unable to survive at certain temperatures; a host may be displaced by human activity; and a vector may be unable to reproduce without access to standing water. The field of geospatial information systems (GIS) provides a large amount of data on such environmental and anthropological factors [29]. Environmental niche modelling utilises this geospatial data to identify relationships between the presence of a pathogen, host or vector and the environments in which they have been observed, allowing for prediction of the suitability for transmission of a vector-borne zoonosis such as *P. knowlesi* across a geographic area of interest [30].

In 2015, Shearer and colleagues applied a niche modelling approach to produce the first predictive map of *P. knowlesi* malaria risk across Southeast Asia [31]. This map provided an initial evidence base for identifying areas where disease surveillance and epidemiological investigations would be most informative to improve understanding of *P. knowlesi* malaria risk. Since the publication of the 2015 occurrence database and risk map, the volume of *P. knowlesi* data has increased across Southeast Asia, with this including detections of the pathogen in new locations. As new data accrues, it is important to update risk predictions to ensure that the most up-to-date evidence is available to public health researchers, practitioners, and policymakers. Furthermore, since 2015, studies providing evidence of the importance of

deforestation in the risk of *P. knowlesi* malaria have been published, and novel datasets characterising spatial and temporal variation in land use patterns have become available.

In this study, we present updates to the *P. knowlesi* infection database and risk map produced in 2015 [31]. We perform a comprehensive review of the literature from October 2015 through to March 2020 to produce a consolidated database of *P. knowlesi* infection occurrences across Southeast Asia. By combining this occurrence dataset with data on a range of environmental covariates using a niche modelling framework, we produce updated predictions of relative suitability for *P. knowlesi* transmission to humans at fine-scale across Southeast Asia. We compare the outputs of our model to those from the 2015 model.

## Methods

### Infection data

The infection occurrence database is a listing of reported locations of *P. knowlesi* infections in either humans, macaques or mosquitoes. The infection occurrences used in the 2015 analysis were extracted from literature published up to October 2015. In order to identify new occurrences, we searched the 'Web of Science' database on March 2nd 2020, using the keywords "knowlesi" *or* "monkey malaria" and filtered for results published after October 2015 (Fig 1A). Following the exclusion of laboratory studies, we extracted infection occurrence records from publications which utilised validated *P. knowlesi*-specific diagnostics (i.e. semi-nested PCR or a combination of microscopy and molecular techniques, as in the 2015 review [31]). The data collection protocol used was the same as in the 2015 analysis and further detail can be found therein [31]. We combined the collected infection occurrences produced by the current study ($n = 264$, Fig 1) with those identified in the 2015 analysis ($n = 260$).

Each location in the infection occurrence database could either take the form of a point or a polygon record. We created point records where the likely exposure site was reported with enough precision that it could be assigned to a $5 \times 5$ km grid cell. Where this level of precision was not available, we created *polygonal* records, assigning the likely exposure site to a region bounded by a polygon (Fig 1B). We created these polygons as either administrative level 1 (the first subdivision below national, e.g. state or province) or administrative level 2 (the second subdivision below national, e.g. district or regency) then disaggregated these polygons onto to the $5 \times 5$ km grid for model fitting, prediction and evaluation.

Prior to model fitting and evaluation, we excluded nine records which spanned an area greater than 1,000 grid cells (approximately 25,000 km$^2$). These records were unlikely to affect results given that each had substantial overlap with other more precise spatial records.

### Covariate data

The infection risk model incorporated 20 environmental covariates (Table 1), each a $5 \times 5$ km gridded raster covering Southeast Asia. Of these 20 covariates, we treated 14 as time-varying with an annual resolution, allowing the model to associate each infection occurrence records with covariate values corresponding to the year the infection was recorded, capturing the variation of risk factors over time. Data for these annually-varying covariates were available for each year from 2001 to 2019, extending upon the coverage of the 2015 model (which covered 2001 to 2015). We assigned five samples which were collected before the year 2001 covariate values for the year 2001.

While tasseled-cap values (transformed Landsat imagery which can help differentiate areas of vegetation and urbanisation) and human population density were included as synoptic (static) variables in the 2015 model, in this work we incorporated them as temporally-varying covariates. The 2015 model incorporated an urban accessibility metric which defined the travel

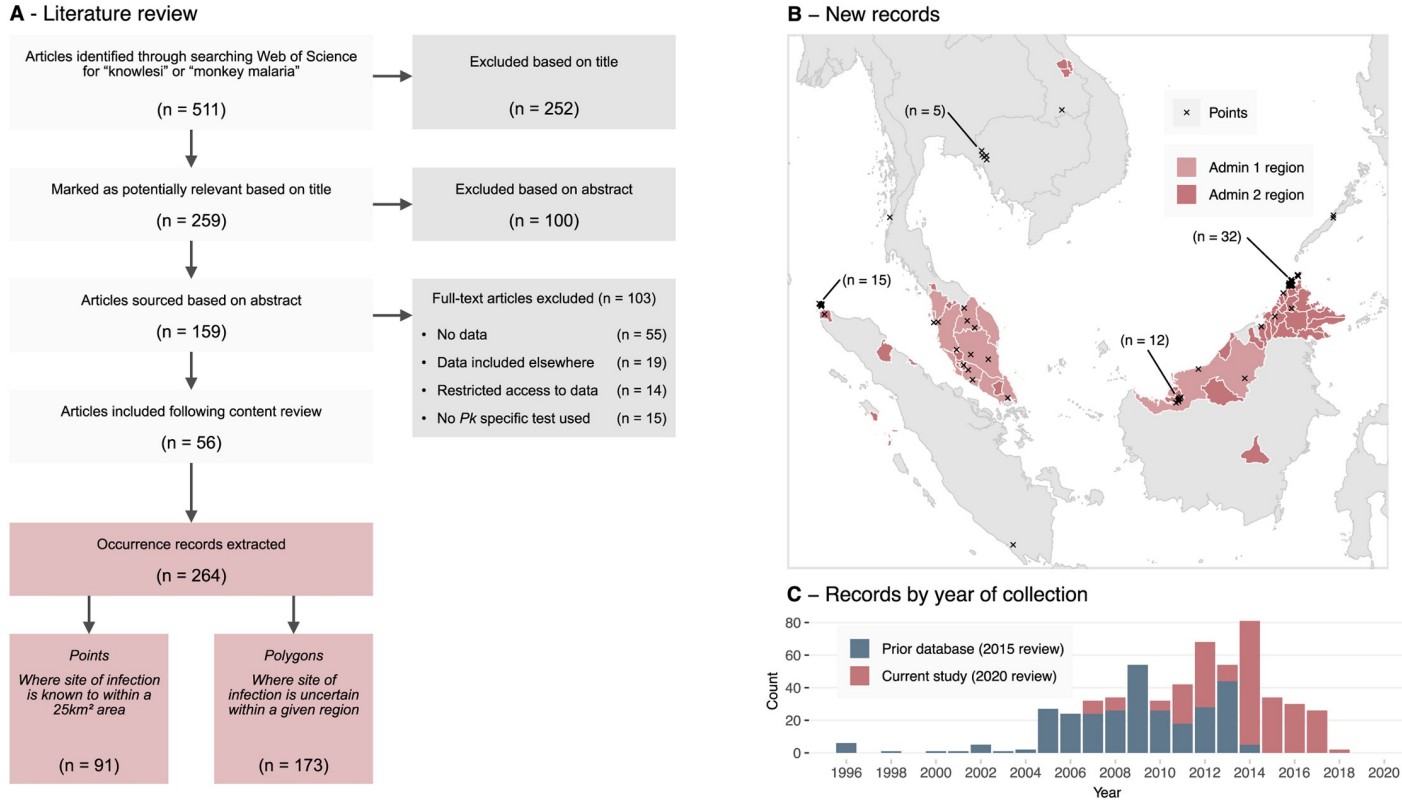

**Fig 1. The results of the March 2020 literature review.** A: Study and sample selection process for the 2020 infection occurrence database. Records were produced via a literature review which was performed on March 2nd 2020, filtering for publications released after October 2015. **B**: Newly extracted point and polygon occurrence records across Southeast Asia by spatial type. Admin 1 regions are the first subdivision below national, e.g. state or province. Admin 2 regions are the second subdivision below national, e.g. district or regency. **C**: The number of occurrence samples in each occurrence database by the year the sample was collected. Administrative boundary base maps sourced from the Malaria Atlas Project (CC BY 3.0, [32]) and international boundaries from the US Department of State Large Scale International Boundaries dataset (public domain, [33]).

time to the nearest city of 50,000 people or more by land- or water-based travel in the year 2000 [34]. Here, we instead used the healthcare accessibility surface—a modelled measure of travel time to the nearest healthcare facility produced by the Malaria Atlas Project which used data up to mid-2019 [35]—as a measure of urban accessibility.

We replaced the intact and disturbed forest coverage layers used in the 2015 model with covariates that better captured the temporal and spatial dynamics of forest change in Southeast Asia. The forest coverage data sets used in the 2015 model were derived from the Intact Forest Landscapes project, which utilised a strict, manually assessed criteria for defining intact versus disturbed forest [31,36,37]. However, the temporal resolution of this dataset is low, with data only available for four distinct years (2000, 2013, 2016 and 2020). We chose instead to utilise data provided through the Global Forest Change project, which provides annual data on tree coverage over the last 20 years on forest presence at the resolution of 1 arc-second (roughly 30 m) [38].

We aggregated the Global Forest Change dataset up to the 5 × 5 km grid over the Southeast Asia study region through the calculation of both a tree coverage and a tree loss metric. We defined tree loss to be the proportion of forest area lost within each 5 × 5 km cell for each study year. Similarly, we defined tree coverage as the proportion of land where forest coverage was present at the beginning of the Global Forest Change data period and where no

**Table 1. The set of raster covariate datasets used in model fitting and prediction.** Differences in raster datasets between this work and those used in the 2015 P. knowlesi risk model appear in bold. STRM: Shuttle Radar Topography Mission, MODIS: Moderate Resolution Imaging Spectroradiometer, IGBP: International Geosphere-Biosphere Programme.

| Name | Description | Temporal resolution |
|---|---|---|
| *Macaca fascicularis* suitability | Modelled suitability of inhabitation by macaques of species *M. fascicularis* [37]. | Synoptic |
| *Macaca nemestrina* suitability | Modelled suitability of inhabitation by macaques of species *M. nemestrina* [37]. | Synoptic |
| *Anopheles* Leucosphyrus Group suitability | Modelled suitability of inhabitation by mosquitoes of the *Anopheles* Leucosphyrus Group [37]. | Synoptic |
| SRTM elevation | Mean elevation [39] | Synoptic |
| Tasseled cap wetness s.d. | Tasseled-cap transformed MODIS data [40, 41]. **Now treated as temporally-varying**. | Annual |
| Tasseled cap wetness mean | " " " | Annual |
| Tasseled cap brightness s.d. | " " " | Annual |
| *Plasmodium falciparum* temperature suitability | Modelled temperature suitability index for *P. falciparum* transmission used as proxy for suitability of *P. knowlesi* [42]. | Synoptic |
| Forest loss | Proportion of land where forest coverage has been lost in a given year [38]. **Replaced the disturbed forest dataset**. | Annual |
| Forest coverage | Proportion of land with forest coverage present in a given year [38]. **Replaced the intact forest dataset**. | Annual |
| Healthcare accessibility | Modelled duration travel time to the nearest healthcare facility [35]. **Replaced the urban accessibility dataset**. | Synoptic |
| WorldPop human population | Mean human population density [43, 44]. **Now treated as temporally-varying**. | Annual |
| Open shrublands | Proportion of land with given land classification [45]. | Annual |
| Woody savannas | " " " | Annual |
| Savannas | " " " | Annual |
| Grasslands | " " " | Annual |
| Permanent wetlands | " " " | Annual |
| Croplands | " " " | Annual |
| Cropland/natural vegetation mosaic | " " " | Annual |
| Urban and built up | " " " | Annual |

subsequent loss was recorded up until each study year. As the Global Forest Change project has not calculated forest gain past the year 2012, we were not able to include any possible increase in forest coverage.

## Model fitting

We utilised a bootstrapped boosted regression tree modelling framework to characterise relationships between a region's environment and the occurrence of *P. knowlesi* transmission. Regression trees produce an approximation of some latent function (e.g. the probability of a *P. knowlesi* infection occurring) by recursively splitting across potential predictor variables (e.g. environmental covariates). The points at which these splits occur and the value assigned across each split region are selected such that the error between the regression tree and the observations is minimised [46]. Boosted regression trees extend upon the regression tree framework by producing a large number of trees and combining them in an ensemble (a process known as boosting) such that they better approximate the latent function [47]. Boosted regression trees are able to fit complex nonlinear responses including high-dimensional interactions between explanatory variables due to their hierarchical tree structure and have been shown to exhibit high predictive accuracy [48]. Finally, bootstrapping of the boosted regression tree process can be performed, allowing for uncertainty in the output to be estimated [49].

When applied to presence-absence data (such as from a systematic survey), niche models generally use a binomial likelihood to represent the probability of a species being present at a given location. Where most of the data available for modelling are presence-only, as is the case for *P. knowlesi* malaria, it is common practice in niche modelling to supplement occurrence records with "background" points to represent areas where the species or disease has not been reported [48]. A variety of approaches have been employed to select background points, including sampling to ensure that their spatial distribution emulates the sampling bias in the presence records [50].

Most *P. knowlesi* occurrences to date are recorded in Malaysia, Brunei and Singapore, with all three of these countries having eliminated the human malaria species (*e.g.*, *P. vivax* and *P. falciparum*), such that that *P. knowlesi* is routinely considered a potential cause of malaria cases. Outside of these countries, surveillance for *P. knowlesi* is limited and infection records are sparse. As per the 2015 study, the goal of our niche modelling analysis is to predict broadly into the under-sampled regions outside of Malaysia, Brunei, and Singapore, using a model fit to data from within these three countries (i.e. the model training region, Fig 2A) where we can account for reporting bias through the selection of background points. Data from outside of these three countries formed the evaluation dataset (i.e. the model evaluation region, Fig 2B), which we used to assess the model's predictive ability outside of the training region.

Background points were produced as in the 2015 analysis [31]. To produce background points for the human records, we sampled points across the training region weighted by human population density [43], under the assumption that human *P. knowlesi* infections would be more likely to be detected within more populous areas. Background points for mosquito records were similarly produced with sampling weighted by human population density, under the assumption that the locations of mosquito infection studies would be selected based on the presence of human *P. knowlesi* cases. To produce macaque background records, we sampled points from a survey of macaques and other mammals [37], as we expected this survey to have similar sampling bias to that of macaque *P. knowlesi* infection records. As in the 2015 model [31], the geographic distribution of *Macaca leonina*—a putative host species of *P. knowlesi* which was only classified as a species distinct from *Macaca nemestrina* in 2001 [51]—has not been included as an explanatory covariate in model fitting as the species is not found in the model training region.

To produce each bootstrap we performed sampling with replacement across each of the combined occurrence polygons, occurrence point records and background points, using occurrence records present in the training region of Malaysia, Brunei and Singapore. We constrained this sampling so that at least 10 presence and 10 background points were present within each bootstrap. Each polygon occurrence record was then reduced to a single point location; this was achieved by selecting a point at random uniformly across each polygon for each bootstrap. As areas of overlapping polygons therefore have a greater probability of a point being sampled, we present the density of overlapping polygons in Fig 2A. For each bootstrap, we assigned weights to sampled points such that the sum of weights for presence points was equal to the sum of weights for the background points, and environmental values were assigned to each point from the set of covariate rasters corresponding to the spatial location and year the sample was recorded. We produced a covariate for host species, indicating if the sample was collected from a human, a mosquito or a macaque. We repeated this process to produce 500 bootstrapped datasets.

For each bootstrapped dataset, we fit boosted regression trees using the gbm3 and seegSDM packages. Hyperparameters for model fitting were unchanged from the defaults provided by seegSDM version 0.1–9 (initial trees = 10, learning rate/shrinkage = 0.005, tree complexity = 4, maximum trees = 10,000). We produced predictions across each of the 500 bootstrapped

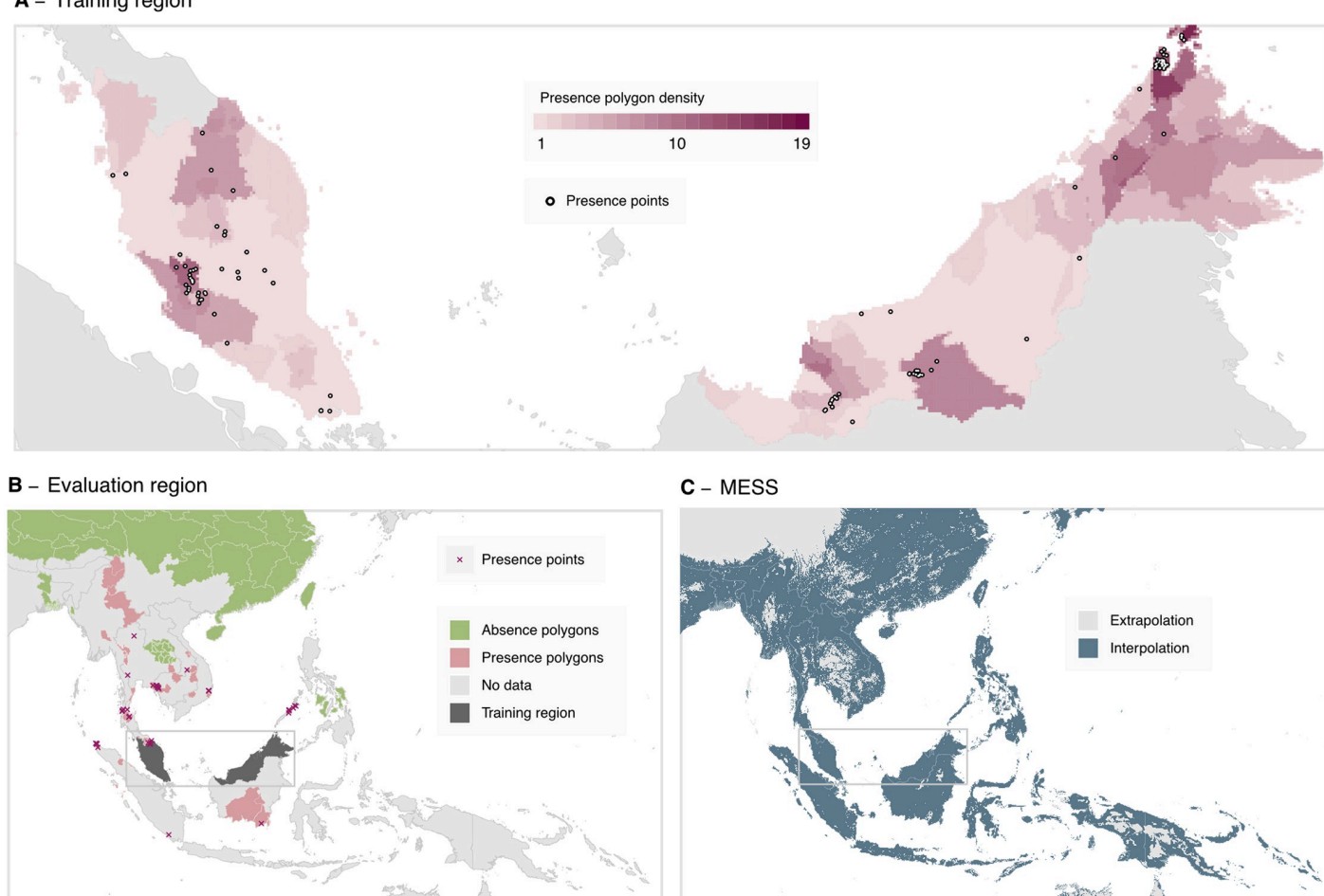

**Fig 2. Data included for modelling across the training and evaluation regions and corresponding multivariate environmental similarity surface (MESS). A**: The data-set of occurrence points and polygons used for fitting the boosted regression tree model across the model training region of Malaysia, Brunei, and Singapore. Presence polygons are displayed as the number of polygons covering each given pixel, with this density being proportional to the probability distribution of points sampled from the polygons for each bootstrap. **B**: The presence and absence records used in the model evaluation process, across the evaluation region of Southeast Asia excluding Malaysia, Brunei and Singapore. **C**: Multivariate environmental similarity surface (MESS) for the model, where areas shaded in light grey indicate that at least one covariate value at that point is outside the range of values within the training data (extrapolation). Administrative boundary base maps sourced from the Malaria Atlas Project (CC BY 3.0, [32]) and international boundaries from the US Department of State Large Scale International Boundaries dataset (public domain, [33]).

models, with summary statistics including mean, variance, and interquartile range calculated for each $5 \times 5$ km grid cell across Southeast Asia (Figure A in S1 Text). Predictions were made using covariate data corresponding to 2019, the most recent year available. As in the 2015 model, we restricted predictions to areas within the range of macaque and mosquito species known to be required for zoonotic transmission of *P. knowlesi* (i.e. the overlap in range maps of at least one reservoir and one vector species), using predicted species extent maps previously reported [37].

We produced a multivariate environment similarity surface (MESS) map (Fig 2C), indicating geographic areas where the value of at least one environmental covariate was outside the range of values present in the training data (i.e. the model is extrapolating) or vice-versa [52].

Prediction results for each bootstrapped model, rasters of summary statistics, the code used to produce results, and the updated occurrence database have been made available at osf.io/k5bsa (DOI 10.17605/OSF.IO/K5BSA).

## Model evaluation

We evaluated the model's predictive performance by calculating the area under the curve (AUC) metric across both the training and evaluation datasets. For the training dataset, we estimated a 10-fold cross-validated AUC throughout the tree count optimisation process, and reported the training AUC for each bootstrap as that of the optimal model selected. Across the evaluation dataset, we calculated AUC across each bootstrapped model, with pairwise distance selection of samples performed to avoid spatial sorting bias [53].

We calculated covariate relative influence scores for each bootstrapped model, representing the number of times a variable is selected for regression tree splitting, weighted by the squared improvement to the model as a result of each split and averaged over all trees [54]. We summarised these scores across the models as means and 95% confidence intervals, with mean values also being used to rank the relative covariate importance. We further calculated accumulated local effect (ALE) scores to describe the average effect of a covariate on the prediction value across the range of each covariate. The ALE score achieves this by identifying how the model prediction changes in response to small changes in the covariate of interest while all other covariates are kept constant, allowing for the effects of covariates to be identified even when the covariates may be highly correlated [55].

## Results

### Infection data

The literature review of articles including data on *P. knowlesi* infection occurrences published between October 2015 and March 2020 returned 511 candidate articles. Following a review of titles and abstracts, 159 articles were deemed likely to contain data for extraction, and 56 articles were identified as meeting the final criteria (Fig 1A). From these 56 articles, 264 occurrences of *P. knowlesi* were extracted, with 91 (34%) being assigned a point record type and 173 (66%) being assigned a polygon record type. Of the 264 extracted records, 241 (91%) were infections identified in humans, with only 14 in macaques and nine in mosquitoes (Table A in S1 Text). The number of records by year of sample collection was greatest in 2014 with 80 records across 14 publications (Fig 1C).

A majority of records added to the 2015 database were collected in Malaysia (*n* = 201, 76%). Within Malaysia, the spatial distribution of records was highly heterogeneous (Fig 1, Table A in S1 Text), with 127 polygon records assigned to the region of Sabah in contrast to the three records identified in the capital region of Kuala Lumpur. Malaysia was also the location of eight of the nine observed infected mosquitoes in the dataset, consistent with the greater sampling effort within the country [19].

Our literature search reveals that more infection occurrences from Indonesia have been reported since 2015, comprising 17% (*n* = 45) of the new presence records (where the prior 2015 literature search identified only five infections within the country). These records are the result of a small number of high-quality surveys and case reports from Aceh [56] and North Sumatra [57]. The literature review dataset contains three records from Laos, where the first confirmed human *P. knowlesi* infection was reported in 2016 [58].

In combination with the 260 occurrences used in the 2015 analysis, the total number of infection occurrences used in model fitting and evaluation was 524. Of these, 396 were within

the training region of Malaysia, Brunei and Singapore, with the remaining 128 located elsewhere in Southeast Asia (Table A in S1 Text).

## Transmission suitability model output

The mean and standard deviation of predicted *P. knowlesi* transmission suitability (a relative measure of the potential risk of *P. knowlesi* transmission to humans) across at-risk areas of Southeast Asia is presented in Fig 3. Predictions were produced using covariate raster datasets as of 2019, representing our most up-to-date estimate of transmission suitability across the region. Further summary statistics of transmission suitability are presented in Figure A in S1 Text.

The map of *P. knowlesi* transmission suitability (Fig 3A) shows highly heterogeneous levels of predicted risk across Southeast Asia. On the island of Borneo, all areas other than lower-lying coastal regions are expected to have a relatively high risk of *P. knowlesi* transmission. Other more sparsely distributed areas of relatively high risk are predicted in Indonesia within the provinces of Sumatra and West Nusa Tenggara. Peninsula Malaysia is predicted to have inland areas of high transmission risk. Thailand, Laos, Cambodia, Vietnam, Myanmar and the Philippine island of Luzon have smaller, localised areas of high predicted risk, with greater uncertainty in these predictions (Fig 3B) as a result of environmental differences to the model training region of Malaysia, Brunei and Singapore.

Within the training region, a mean area under the curve (AUC) of 0.81 was produced across the 500 bootstrapped models with a standard error of 0.001. For the evaluation region, the mean AUC was found to be 0.75 with a standard error of 0.003.

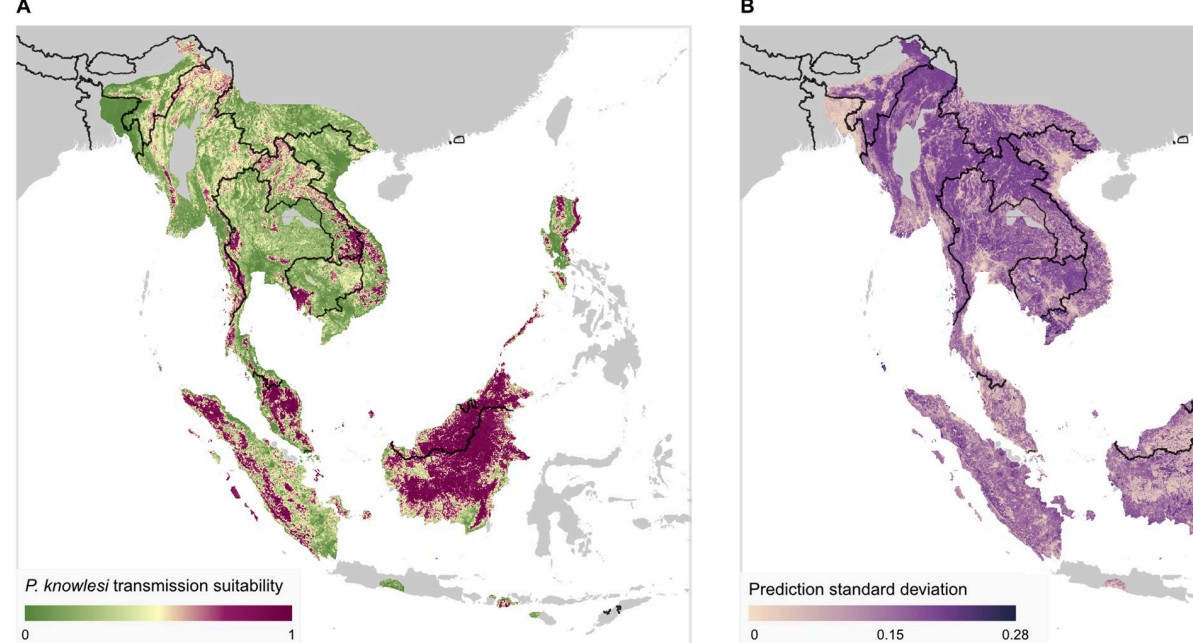

**Fig 3. Predicted *P. knowlesi* transmission suitability across Southeast Asia. A:** Modelled transmission suitability mean over Southeast Asia across the 500 bootstraps. Results are displayed only where an area is within the range of both a vector and reservoir species necessary for transmission (see Methods), regions outside of this range (displayed as grey) are considered to be very low risk for *P. knowlesi* transmission. Transmission suitability is a relative measure of the risk of *P. knowlesi* transmission from known reservoir species (via vector species) to humans. **B:** Standard deviation of the predicted transmission suitability across the 500 bootstraps. Administrative boundary base maps sourced from the Malaria Atlas Project (CC BY 3.0, [32]) and international boundaries from the US Department of State Large Scale International Boundaries dataset (public domain, [33]).

These values indicate a high degree of predictive performance.

Examining predictions within the evaluation region of the model (Southeast Asia excluding Malaysia, Brunei and Singapore), we may qualitatively assess the model's predictive performance. Regions with both a high modelled transmission suitability and previously identified occurrence samples of *P. knowlesi*—indicative of good model sensitivity—include the Aceh province of Sumatra island in Indonesia, the Koh Kong province in southern Cambodia and the Mimaropa region of the Philippines (Figure B in S1 Text, panel A). We also see that there are a substantial number of regions where the model predicts high transmission suitability where *P. knowlesi* occurrence has not previously been identified as of the 2020 literature review (i.e. omission errors [59]). This includes the province of West Nusa Tenggara in Indonesia (Figure B in S1 Text, panel A). Such predictions may be suggestive of a lack of surveillance in these regions, or that an environment is conducive to transmission but currently lacking widespread occurrence of a necessary vector or host species.

We additionally examine the performance of the predicted *P. knowlesi* transmission risk map presented by the 2015 analysis against the occurrence data collected in our literature review (Figure G in S1 Text). We find that, qualitatively speaking, the performance of the 2015 analysis in predicting the presence of infection occurrences published in the literature between October 2015 and March 2020 was good.

The covariate of human population density was found to have the highest ranked relative influence for the majority (496/500, 99.2%) of the bootstrapped models, closely followed by that of healthcare accessibility. Mean and 95% confidence intervals of relative influence scores across bootstraps are presented in Figure E in S1 Text. The new covariates of tree coverage and forest loss were found to be highly influential; out of 21 covariates (20 environmental covariates and the species covariate), the median rank for tree coverage was 5 (95% confidence interval: 2–11), and for forest loss was 10 (95% confidence interval: 6–15). Plots of the accumulated local effects (ALE) describing the influence of each continuous covariate across the covariate's range are presented in Figure F in S1 Text.

## Discussion

In this study, we utilised an environmental niche modelling approach to predict the relative suitability for *P. knowlesi* transmission to humans across Southeast Asia. We extended a previous analysis that incorporated data up to 2015 [31] by adding infection and environmental data up to 2020, and improving the utilisation of data on land use patterns. Through a review of literature published between October 2015 and March 2020, we identified 264 published occurrences of *P. knowlesi*. This resulted in a total of 524 records being utilised in model fitting and evaluation for the current study. As changes in *P. knowlesi* transmission risk may be expected where substantial amounts of deforestation have occurred [24–27], we now capture this in the model by deriving annual forest loss and coverage datasets. We predict that the distribution of *P. knowlesi* risk is highly heterogeneous across Southeast Asia, with the largest areas of predicted risk in Malaysia and Indonesia, and smaller, localised regions of high risk predicted in the Greater Mekong Subregion, The Philippines and Northeast India.

Our analysis can help to guide the prioritisation of locations for future sampling and surveillance for P. knowlesi malaria by highlighting areas of high predicted risk that may have been under-sampled. Since the publication of the 2015 analysis, there has been no change to the World Health Organization's malaria elimination status of any country believed to be at risk for indigenous *P. knowlesi* transmission [60]. However, within the Greater Mekong Subregion of Cambodia, Myanmar, Thailand, Laos and Vietnam, substantial declines have been observed in the total number of reported malaria cases as of 2021 [61]. The 2015 analysis

noted that Laos, Myanmar, Thailand and Vietnam were likely high-value sites for future sampling efforts [31] and our literature search revealed only a small number of additional *P. knowlesi* occurrences in these countries as of 2020 (Fig 1, Table A in S1 Text). Our current analysis predicts localised areas of moderate-to-high relative transmission risk in this region (Fig 3), suggesting an ongoing need for surveillance of *P. knowlesi* malaria. In contrast, Palawan in The Philippines was also highlighted as a target for future sampling efforts in the 2015 analysis [31], with new occurrence records confirming the presence of *P. knowlesi* in this region [62].

Indonesia has a stated goal to eliminate malaria by 2030 [63,64], and may be on track given that a majority of administrative regions have declared elimination [65]. However, the presence of *P. knowlesi* across the country presents a serious challenge to these efforts. In March 2022, the WHO Malaria Policy Advisory Group (MPAG) concluded that certification of malaria elimination status should only occur where the risk of *P. knowlesi* was 'negligible', i.e. below some low threshold of annual incidence [60,66]–a requirement that has already prevented Malaysia from receiving elimination certification [67]. Given this requirement, continued surveillance and mitigation of *P. knowlesi* throughout at-risk regions of Indonesia will be important. Between the period of 2015 and 2020, a small number of studies have identified substantial numbers of *P. knowlesi* infections within Indonesia [68–70], particularly within northern Sumatra [56,71,72], a region identified as a valuable target for surveillance effort in the 2015 model [31]. Despite this, the Indonesian region of Kalimantan on the island of Borneo still has a relative scarcity of occurrence data given its high predicted transmission suitability and the number of P. knowlesi cases reported in adjacent areas of Malaysia.

Our map of *P. knowlesi* transmission risk may also help to quantitatively guide site selection for public health surveillance or intervention. For example, surveillance sampling could be concentrated in regions where the model predicts a high transmission suitability but with a high variance, such that the understanding of the geographical distribution of the disease is maximised for the least effort and that the uncertainty in these regions could be reduced in future risk mapping outputs. If value was instead placed on maximising the probability of identifying cases of *P. knowlesi*, sampling could be concentrated where a high transmission suitability is accompanied by lower variance. Efficient deployment of sampling resources could be achieved by combining the modelling outputs with constraints; for example, sites where access would require a prohibitive amount of travel time could be excluded [73].

Updating a model of risk—as we have performed here for *P. knowlesi* transmission suitability—raises key questions regarding when and why such an update should be performed. We produced the update in response to two primary factors: the accumulation of further *P. knowlesi* infection occurrence data since the publication of the previous mapping study, which was expected to improve estimate precision when incorporated into the model; and changes in land cover across Southeast Asia, such as deforestation, which were suspected to have caused changes in the underlying distribution of transmission risk. For our predictions (with covariate data as of 2019 and occurrence data up to 2018) to be sufficient for future sampling efforts, the change in the underlying distribution of risk over time should be minimal. However, identifying if such changes have occurred is particularly difficult for *P. knowlesi* given the nature of data collection for the pathogen. Whereas studies of the human malaria species are able to isolate the effect of a changing risk distribution through the use of data which has been collected in a systematic manner [74], the data available for our study largely comprises infection occurrences identified by localised prospective sampling and passive surveillance. Despite these difficulties, we note that the good predictive performance of the 2015 analysis (Figure G in S1 Text) provides reason to believe that our static estimates are sufficient to inform future sampling efforts.

As niche modelling frameworks are correlative, the secondary results described in this work should be interpreted with care. The relative influence scores (Figure E in S1 Text) and the accumulated local effect plots (Figure F in S1 Text) may provide insight into risk factors for *P. knowlesi* transmission. However, these results do not provide evidence for causal relationships, which would instead be more appropriately identified through studies utilising a causal inference framework. For example, the covariate of healthcare accessibility, which ranks highly according to relative influence scores (Figure E in S1 Text), could capture a direct causal effect on the risk of being diagnosed with *P. knowlesi* (e.g. likelihood that someone is identified as having a *P. knowlesi* infection increasing with access to healthcare) or may simply be confounded by a common variable (e.g. likelihood of acquiring a *P. knowlesi* infection increasing for those who work at plantations, confounded by such plantations occurring in areas of lower healthcare accessibility).

Our model predicts the relative suitability for *P. knowlesi* transmission, not the prevalence of infection nor the incidence of cases (which would require different input data that are not widely available for *P. knowlesi* malaria). While transmission suitability is a useful metric for prioritising locations for future P. knowlesi surveys, the absolute values are specific to the input data and model parameterisation, and we therefore cannot directly compare absolute values produced by the model presented here and those from the model developed in 2015. Although we expect that the transmission suitability prediction produced by either of the models should be qualitatively related to the underlying 'true' risk of *P. knowlesi* infection, little can be said of this relationship other than that it is expected to be monotonic under the assumption that the background data points are biased in the same manner as the presence data [50]. This means, for instance, that any differences between the models that could arise as a result of dilation in this relationship (such as the upwards dilation observed in Figure D in S1 Text) cannot be taken alone as indicating a change in underlying transmission suitability.

Noting the limitations in these comparisons, we find that the predictions in our work and the 2015 model broadly align, though with clear differences in the local spatial variation of the prediction surface (Figure C in S1 Text, Figure D in S1 Text, panel A). As an example, on the island of Borneo our predictions form a smooth region of high predicted risk, whereas in the 2015 model predictions over the same area varied substantially at a small spatial scale; this pattern is repeated similarly elsewhere across Southeast Asia [31]. In countries such as Laos, Myanmar and Vietnam, we predict overall a lower transmission suitability than those presented in the 2015 model, though within these countries we continue to predict small areas of high transmission risk. Comparing the overall distributions of predicted transmission suitability between the 2015 and 2020 models shows that our new predictions produce a more highly contrasting bimodal distribution of risk compared to that produced by the 2015 model [31] (Figure D in S1 Text, panel A). Identifying potential reasons for such changes in model output is also difficult. Such an analysis would require a systematic examination of differences in model structure, covariate data and occurrence data, and, given the correlative nature of the model, would not necessarily be expected to provide insight into the mechanisms of *P. knowlesi* risk.

The temperature suitability index covariate used in the model attempts to describe the effect of temperature on the basic reproduction number for some combination of malaria parasites and mosquito vectors. As data on the incubation periods for *P. knowlesi* under differing temperatures and mosquito hosts is currently unavailable, no suitability index for the species can currently be produced. In this work, we instead utilise a proxy in the form of a suitability index for *P. falciparum* [42]. Even if this proxy does not itself accurately capture mechanistic limits on *P. knowlesi* reproduction, it is not immediately obvious what bias this would introduce into the results, if any, as the boosted regression tree model may still infer suitability under some

transformation of the index. There is clear value in further laboratory research on the reproduction of *P. knowlesi* under different temperatures that could inform a species-specific suitability index.

It is believed that workers involved in the development and cultivation of oil palm plantations are at greater risk for developing *P. knowlesi* infection given their proximity to *P. knowlesi* vector and reservoir species [75]. However, we were unable to include this as a covariate in our model as there is currently no published dataset of palm oil plantations with complete coverage across the Southeast Asia region.

In our approach, we implicitly assume that under-ascertainment of *P. knowlesi* infections in humans due to asymptomatic/submicroscopic or spontaneously resolving disease (see [9–11]) has a uniform effect geographically. If the data used for this study were biased by such under-ascertainment which was not uniform across space (e.g. due to differing levels of immunity between regions), this would be expected to in turn bias our predicted transmission suitability downwards in environments similar to those where under-ascertainment were occurring. Further research on any potential spatial association of asymptomatic or submicroscopic human *P. knowlesi* infection would be of high value in further refining estimates of the spatial distribution of transmission of the parasite.

Annual data is not available for some of the covariates used in the model where the underlying phenomena may be expected to change over time; the covariates of reservoir/vector species distribution and temperature suitability are dependent upon variables such as climate or land cover, and the covariate of healthcare accessibility is dependent upon changes in transportation infrastructure and locations of healthcare sites. In lieu of available data on change in these covariates over time they are instead assumed to be constant. In effect, this means that the modelled species distributions as of 2014, temperature suitability index for *P. falciparum* as of 2010 and healthcare accessibility as of 2019 are all assumed constant over the years 2001 to 2019. Future modelling efforts could be improved by considering the change in these covariates over time.

Our map of *P. knowlesi* transmission suitability predict high *P. knowlesi* disease risk across broad areas of Southeast Asia, with large regions of high predicted *P. knowlesi* risk that have not yet been sampled for the pathogen. Our work demonstrates the importance of continued surveillance and prospective sampling of the pathogen, especially in regions where malaria elimination is currently being pursued.

## Supporting information

**S1 Text. SupplementaryMaterials—Updating estimates of Plasmodium knowlesi malaria risk in response to changing land use patterns across Southeast Asia. Figure A in S1 Text. Summary statistics for modelled transmission suitability across Southeast Asia, calculated across the set of 500 bootstraps.** Results are only displayed where an area is in the range of both a vector and reservoir species necessary for transmission (see Methods). Administrative boundary base maps sourced from the Malaria Atlas Project (CC BY 3.0, [32]) and international boundaries from the US Department of State Large Scale International Boundaries dataset (public domain, [33]). **Figure B in S1 Text. Predicted mean transmission suitability with overlay of infection occurrence data across the evaluation region.** Results are only displayed where an area is in the range of both a vector and reservoir species necessary for transmission (see Methods). **A**: Predicted transmission suitability with infection occurrence polygons and points in blue. **B**: Predicted transmission suitability with infection absence polygons in blue. Administrative boundary base maps sourced from the Malaria Atlas Project (CC BY 3.0, [32]) and international boundaries from the US Department of State Large Scale International Boundaries dataset (public domain, [33]). **Figure C in S1 Text. Comparison of the modelled**

**mean transmission suitability value between the current work and the 2015 model.** Model predictions are presented from the current work with data as of 2020 (**A**), and the predictions presented in the 2015 model [31] (**B**). Note that the absolute value of predictions are not necessarily comparable given differences in model specification and training data. Results are only displayed where an area is in the range of both a vector and reservoir species necessary for transmission (see Methods). Administrative boundary base maps sourced from the Malaria Atlas Project (CC BY 3.0, [32]) and international boundaries from the US Department of State Large Scale International Boundaries dataset (public domain, [33]). **Figure D in S1 Text. Changes in the distribution of predicted transmission suitability between the current study and the 2015 model.** Histograms (**A**) and quantile-quantile plot (**B**) comparing the distributions of mean predicted transmission suitability for the 2015 and 2020 models of *P. knowlesi* transmission risk. Histograms are presented on a relative x-axis (ranging from minimal to maximal predicted mean risk), with quartiles of predicted risk displayed as dashed vertical lines. **Figure E in S1 Text. Relative influence scores for each covariate.** Scores are calculated for each bootstrap, with points and lines representing median values and 95% confidence intervals respectively. **Figure F in S1 Text. Accumulated local effect (ALE) plots for each covariate.** ALE indicates the mean effect of changing a covariate's value upon the prediction (on logistic scale) across the range of that covariate. The ALE values are calculated for each bootstrap, with the median value, 50% and 95% confidence intervals presented as lines, darker shaded regions and lighter shaded regions respectively. **Figure G in S1 Text. Comparison of predicted transmission risk as presented in the 2016 work and the occurrence data collected in the 2020 literature review.** Results are only displayed where an area is in the range of both a vector and reservoir species necessary for transmission (see Methods). Administrative boundary base maps sourced from the Malaria Atlas Project (CC BY 3.0, [32]) and international boundaries from the US Department of State Large Scale International Boundaries dataset (public domain, [33]). **Table A in S1 Text. The number of human, macaque and mosquito samples in the occurrence database produced by the 2015 literature review.** Samples in Indonesia and Malaysia are shown stratified by region (province, state or territory). Total counts are shown for records from both the 2020 literature review and 2015 literature review.

(PDF)

## Acknowledgments

This research was supported by The University of Melbourne's Research Computing Services and the Petascale Campus Initiative (www.unimelb.edu.au). We thank Dr Timothy William for their support. We would like to also thank the Director General of Health Malaysia for the permission to publish this article.

## Author Contributions

**Conceptualization:** Matthew J. Grigg, Jennifer A. Flegg, David J. Price, Freya M. Shearer.

**Data curation:** Ruarai J. Tobin, Meg K. Tully.

**Formal analysis:** Ruarai J. Tobin, Lucinda E. Harrison.

**Funding acquisition:** Inke N. D. Lubis, Rintis Noviyanti, Nicholas M. Anstey, Giri S. Rajahram, Matthew J. Grigg, Jennifer A. Flegg, David J. Price, Freya M. Shearer.

**Investigation:** Ruarai J. Tobin.

**Methodology:** Ruarai J. Tobin, Lucinda E. Harrison, Jennifer A. Flegg, David J. Price, Freya M. Shearer.

**Project administration:** Inke N. D. Lubis, Rintis Noviyanti, Nicholas M. Anstey, Giri S. Rajahram, Matthew J. Grigg, Jennifer A. Flegg, David J. Price, Freya M. Shearer.

**Resources:** Inke N. D. Lubis, Rintis Noviyanti, Nicholas M. Anstey, Giri S. Rajahram, Matthew J. Grigg, Jennifer A. Flegg, Freya M. Shearer.

**Software:** Ruarai J. Tobin, Lucinda E. Harrison.

**Supervision:** Jennifer A. Flegg, David J. Price, Freya M. Shearer.

**Validation:** Ruarai J. Tobin.

**Visualization:** Ruarai J. Tobin.

**Writing – original draft:** Ruarai J. Tobin, Freya M. Shearer.

**Writing – review & editing:** Ruarai J. Tobin, Lucinda E. Harrison, Meg K. Tully, Inke N. D. Lubis, Rintis Noviyanti, Nicholas M. Anstey, Giri S. Rajahram, Matthew J. Grigg, Jennifer A. Flegg, David J. Price.

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
