## [Decision Letter · Decision Letter 0]

13 Nov 2023

Dear Mr Tobin,

Thank you very much for submitting your manuscript "Updating estimates of *Plasmodium knowlesi* malaria risk in response to changing land use patterns across Southeast Asia" for consideration at PLOS Neglected Tropical Diseases. As with all papers reviewed by the journal, your manuscript was reviewed by members of the editorial board and by several independent reviewers. The reviewers appreciated the attention to an important topic. Based on the reviews, we are likely to accept this manuscript for publication, providing that you modify the manuscript according to the review recommendations. 

Sincerely,

Richard Reithinger

Academic Editor

Charles Jaffe

Section Editor

Reviewer's Responses to Questions

**Key Review Criteria Required for Acceptance?**

**Methods**

-Are the objectives of the study clearly articulated with a clear testable hypothesis stated?

-Is the study design appropriate to address the stated objectives?

-Is the population clearly described and appropriate for the hypothesis being tested?

-Is the sample size sufficient to ensure adequate power to address the hypothesis being tested?

-Were correct statistical analysis used to support conclusions?

-Are there concerns about ethical or regulatory requirements being met?

Reviewer #1: The methods outlined are reasonable and sufficient for the analysis undertaken. The conform to a number of standards in the field. I have no issues with the methodology in the form presented, and while there are possible extensions, I have found that they make little difference to the results, and they are not necessary to undertake in all instances.

Reviewer #2: I agree BRTs are an appropriate tool for this sort of modelling. But given this work is set into context of increasing risk, I think some discussion about the static nature of the presented results [Fig3A] is necessary: is the presented transmission suitability a mean over the time period of data collection, or contemporary suitability? 

Background points [L205]: I think this needs clarification – is the assumption that more populated areas are more likely to report cases for a given incidence, or that incidence is higher and therefore cases are more likely to occur? I would imagine more urban populations would be at lower risk of Knowlesi infection.

Similarly, for the macaque background records [L210/11] – assuming sub-microscopic disease to be uniform geographically needs a citation, ideally.

[L218-220] - what does ‘degraded’ mean in this context? Generally, I don’t think this explanation is very clear to a more general audience, and Figure2A doesn’t particularly help. It is not clear to me how polygon and pixel data have been combined in this framework.

**Results**

-Does the analysis presented match the analysis plan?

-Are the results clearly and completely presented?

-Are the figures (Tables, Images) of sufficient quality for clarity?

Reviewer #1: Given that this is an updating of a prior analysis, I was very surprised to see how little the prior analysis factored into the results. Why was the update required now? Why not a year prior, why not a year later? How good were the old predictions in the context of the new occurrence data? Was incorrect prediction a trigger for re-running? If the objective is for stakeholders to use these maps to make decisions with, demonstrating that prior versions were or were not sufficient to make similar decisions is a critically important perspective to provide. Should I wait until 2030 for the next iteration before I am finally confident that things are stable? We have to make decisions today - how confident can I be in making those decisions with this resource, given the past performance of the prior analysis?

Reviewer #2: The results are coherently presented. 

Transmission suitability [L280]. Please define this metric when it’s introduced, as it means different things to different audiences and it’s not clear quite what is meant until line 304 down the page.

Minor comment: Why a diverging (rather than sequential) colour ramp for Figure 3A?

**Conclusions**

-Are the conclusions supported by the data presented?

-Are the limitations of analysis clearly described?

-Do the authors discuss how these data can be helpful to advance our understanding of the topic under study?

-Is public health relevance addressed?

Reviewer #1: I don't think the main conclusions are too different from the prior modelling exercise, in spite of the new data and covariates. I do not think this is a bad thing - instead this provides a very unique opportunity to retrospectively evaluate the value of the prior exercise in the context of the new data, and justify whether the new methods changes were necessary. Allowing readers to appreciate when, or not, a model is worth re-doing, and how to track that ongoing performance is key. Areas in northern Myanmar are very different in the environmental suitability index score [although direct comparison of this index value with 2015 index values is to be strongly cautioned]. Are any of the guidance from the 2015 paper found to be different in the context of the data and methods upgrades? When should the next assessment be done? Does it require data (if so, in what places?), does it require covariates to change (e.g. climate change shifting things in a way not observed previously?), do we just wait 8 years? What should we be looking out for as concerning enough to prompt a new model run?

Reviewer #2: The conclusions of the paper are supported by the results, and the limitations are clearly described.

Discussion [L378-387]: Are you able to explain why (even speculatively) these changes have occurred since the 2015 work? EG do the covariates look substantially different?

**Editorial and Data Presentation Modifications?**

Reviewer #1: (No Response)

Reviewer #2: Minor comment: Why a diverging (rather than sequential) colour ramp for Figure 3A?

**Summary and General Comments**

Reviewer #1: (No Response)

Reviewer #2: Knowlesi malaria is a growing concern, with recent upticks in south-east Asian countries who are otherwise on the brink of elimination. Although somewhat incremental, I feel this is a helpful addition to the literature, being updates of eight-year-old risk maps including more data and improved covariates. I think some of the explanations could be clearer for non-specialist audiences.

PLOS authors have the option to publish the peer review history of their article (what does this mean?). If published, this will include your full peer review and any attached files.

Reviewer #1: No

Reviewer #2: No

Figure Files:

Data Requirements:

Reproducibility:

References

---

## [Editor Report · Decision Letter 1]

16 Jan 2024

Dear Mr Tobin,

We are pleased to inform you that your manuscript 'Updating estimates of *Plasmodium knowlesi* malaria risk in response to changing land use patterns across Southeast Asia' has been provisionally accepted for publication in PLOS Neglected Tropical Diseases.

Best regards,

Richard Reithinger

Academic Editor

Charles Jaffe

Section Editor

Thank you for addressing the reviewers' comments in full.

---

## [Editor Report · Acceptance letter]

19 Jan 2024

Dear Mr Tobin,

We are delighted to inform you that your manuscript, "Updating estimates of *Plasmodium knowlesi* malaria risk in response to changing land use patterns across Southeast Asia," has been formally accepted for publication in PLOS Neglected Tropical Diseases.

Best regards,

Shaden Kamhawi

co-Editor-in-Chief

Paul Brindley

co-Editor-in-Chief
